# 'When Faith Is Not Enough': Encounters between African Indigenous Religious Practices and Prophetic Pentecostal Movements in Zimbabwe

Molly Manyonganise [1,2]

1 Department of Religious Studies and Philosophy, Faculty of Arts, Culture and Heritage Studies, Zimbabwe Open University, Harare P.O. Box MP1119, Zimbabwe; manyonganisem@zou.ac.zw
2 Department of Religion Studies, Faculty of Theology and Religion, University of Pretoria, Pretoria 0002, South Africa

**Abstract:** African Pentecostalism remains the fastest growing form of Christianity on the African continent. Scholarship on Zimbabwean Pentecostalism has noted how the emergence of New Pentecostal Movements (NPMs), specifically Prophetic Pentecostalism (PP), has increased this growth. Apart from other attracting factors, such as the Holy Spirit, claims of faith healing, deliverance and prophecy, among others, African Pentecostalism is known for its emphasis on faith as a major anchor of any Pentecostal Christian. Hebrews 11, with its emphasis on faith, is, therefore, a central scripture in this Christian tradition. However, the emergence of NPMs at the height of the Zimbabwean crisis from the year 2008 to the present, has challenged Zimbabwean Pentecostal Christians from their sole dependency on faith. The crisis called for much more than faith could stand on its own. Hence, NPMs responded to this need by infusing indigenous religious practices with biblical ones as a way of strengthening believers through the crisis. Prophetic Pentecostal Movements (PPMs) in Zimbabwe introduced touchable objects such as anointed towels, handkerchiefs, wrist bands, stickers, oils and even condoms. While this appears to be sophisticated syncretism, a critical analysis of the practices shows how steeped they are in the African indigenous religious worldview. This article, therefore, seeks to examine the religious encounters between indigenous African religious practices and Pentecostal practices as practiced in the NPMs in Zimbabwe. The focus of this paper is to establish the resilience of indigenous religious practices within a Christian tradition that claims to have totally broken from the past. It further argues that the fast growth of PPMs depends on the 'Christianization' of indigenous religious practices, which are presented to believers as 'purely biblical'. This is largely a desktop research project in which secondary sources were used as sources of data.

**Keywords:** African Pentecostalism; African Traditional Religion(s); Christianity; New Pentecostal Movements; Prophetic Pentecostalism

## 1. Introduction

African Pentecostalism remains the fastest growing form of Christianity on the African continent. Many scholars have engaged in discourses on Pentecostalism on the African continent (see Kalu 2008; Asamoah-Gyadu 2012; Biri 2012, 2020; Chitando et al. 2013; Chitando 2021; Manyonganise 2016, 2020, 2021). Its history has been well-documented from a variety of vantage points. Scholarship focusing on Pentecostalism on the continent have put them into various categories. For example, Yong (2005, p. 18) has three categories, namely, the Classical Pentecostal Movement, the Charismatic Renewal Movement (arising from mainline Protestant, Orthodox and Roman Catholic churches) and the Neo-Charismatic Movements. On the other hand, Togarasei (2018) classifies them into two main categories, namely, Classical and Modern Pentecostal Churches. This study takes note of the various typologies making up Pentecostalism both globally and specifically in Africa, while moving

further to focus on Zimbabwe. Togarasei (2018) provides the history of Pentecostalism in Zimbabwe, which this article will not repeat. The focus of this study is on New Pentecostal Movements in Zimbabwe.

Scholarship on Zimbabwean Pentecostalism has noted that the emergence of New Pentecostal Movements (NPMs), specifically Prophetic Pentecostalism, has increased this growth. Manyonganise (2016, p. 269) notes that the emergence of PPMs in Zimbabwe received a euphoric response. It is important to note that even the new Pentecostal Churches have been differently named. While Manyonganise (2016) calls them Prophetic Pentecostal Movements, Kgatle (2021) refers to them as New Prophetic Churches in his examination of the cultism that is pervasive within the churches, and Biri (2021) calls them Newer Pentecostal Churches. The movements can also be referred to as New Pentecostal Movements. These terms may at times be used interchangeably in this article. Apart from other attracting factors, such as the Holy Spirit, claims of faith healing and deliverance, among others, as shall be shown later, African Pentecostalism is known for its emphasis on faith as a major anchor of any Pentecostal Christian. However, PPMs have also made prophecy central to their theology. Other scholars like Shoko and Chiwara (2013) and Mwandayi (2013) have situated many of the practices and beliefs of PPMs in African Indigenous Religion(s). This article, therefore, seeks to examine the religious encounters between indigenous African religious practices and Pentecostal practices as practiced in Prophetic Pentecostalism in Zimbabwe. Examples are drawn from Emmanuel Makandiwa's United Family International Church and Walter Magaya's Prophetic Healing Deliverance Ministries, as they are currently the most prominent PPMs in Zimbabwe. This is largely a desktop research project, in which secondary sources were used as sources of data. In the next section, I proceed to theorize syncretism.

## 2. Theorizing Syncretism

Theorizing syncretism is no easy task. However, I am convinced that it is necessary to theorize the term before presenting how it is manifesting itself within NPMs in Zimbabwe. The term itself is a contested one, but one that is used to label other people's religions. Werbner (1994, p. 201) argues that syncretism is a social action that is continually contested. However, when it functions across cultures or traditions, it adopts, replicates, or creates religious belief and practices. For him "syncretism is full of controversies some of which emanate from the very concept itself, others about what is pronounced to be the diabolical sin of mixing religious traditions, still others about the authenticity of reconciling religious differences" (Werbner 1994, p. 201). Leopold and Jensen (2014, p. 4) are of the view that syncretism represents elements of struggle in the transmission of religion. In their analysis, "the causes as well as the effects of historical processes that lead to syncretistic formations vary a great deal depending on the historical, cultural, political and social climates". For them, "the use of syncretism in the more theological (mostly Christian) sense has often been used to appoint and control what has been thought of as illegitimate correlations between competing religious movements, traditions or discourses" (2014, p. 8).

Scholarship has alluded to the fact that the labelling of religions as syncretistic is political. Werbner (1994, p. 201) argues that the politics of syncretism is a politics of interpretation and reinterpretation. When I was preparing to write this article, I inquired on possible references to consult with two eminent scholars on religion studies. David Maxwell, who has written widely on African Pentecostalism, with a specific focus on Zimbabwe, responded via email and said this about syncretism: "The key point is that it is a form of religious politics-a term of disapprobation. So, the point is to see who is accusing who of being syncretistic and why? It is about religious boundary maintenance" (Email message, 25 September 2023).

The other scholar, Ezra Chitando, quipped: "is there a religion that is not syncretistic?" (General discussion, 19 September 2023). In concurrence with Chitando, Leopold and Jensen (2014, p. 5) argue that the history of religion confirms that every religion is, in essence, syncretistic. In the same vein, Stewart (2014, p. 275) avers that there are no pure

religious traditions, which, for him, defeats the argument that "syncretism necessarily assumes the existence of ideal pure traditions in contrast to which other traditions are mixed, or syncretic". Stewart further argues that

> The syncretic-ness of all religions may be an unexceptional fact, but pointing this out socially often amounts to an expression of power differentiation and social control. It is a term that has historically been applied to someone else's body of religious practice. The bearers of a given tradition rarely acknowledge that it might be syncretic.

An analysis of Stewart's view brings out two key critical issues. First, that the one labelling another's religion as syncretistic should wield some power over the labelled. Second, that an insider of any religion may never agree that their religion is syncretistic. Hence, only outsiders can label a religion to be syncretistic. In this case, the term carries with it overtones of othering. Hence, Stewart questions the authority of those who label a particular religion to be syncretistic.

While some syncretistic practices are visible, others are very subtle. From Leopold and Jensen's point of view, the majority of syncretistic formations go unnoticed, as they appear as the natural results of interaction (Leopold and Jensen 2014, p. 4). What is important to notice is the function of such syncretistic practices. Stewart (2014, p. 275) opines that syncretism plays a role in directing the invention of traditions or the aggressive dismissal of neighboring traditions. Werbner locates its function within the politics of cultural difference and social identity. In other words, while adopting a foreign religion, some cultural groups may accommodate their own religious beliefs and practices as a way of safeguarding their socio-cultural identity. It is also possible for some religious groups to copy other religions' beliefs and practices. Leopold and Jensen (2014, p. 4) note that new religious movements may incorporate 'borrowed' elements from other religions or secular sources as a way of legitimizing new contemporary values, often disguised as old religious teachings, as a means to contest conventional values. In the Western world, Morazzini (2015) notes that globalization and human migration have fueled syncretistic tendencies.

Within Africa, Stewart (2014) notes that syncretism has become a term of abuse, mostly used to castigate local colonial churches that had burst out from the sphere of mission control and begun to indigenize Christianity. In this case, the use of the term took on a negative sense. Yet these churches sought to fight the cultural imperialism of missionary Christianity and to recover indigenous cultures and values in their traditional religions and cultures denigrated by Christianization and westernization (Leopold and Jensen 2014). Sundkler (1961, p. 55) viewed Zionism in South Africa as a 'nativistic-syncretistic' interpretation of Christianity. Sundkler is credited for his study on Bantu prophets (1948), which illuminated African religiosity and the syncretic merger of African and Western Christianity. Zehner (2009) analyzed the pejorative use of the term 'syncretism'. From his point of view, the term is made judgmental by the assumption that when one religion is either blended, hybridized or contextualized with another, it then represents a deviation from cultural or religious templates that would have been 'pure' were it not for these developments. Zehner (2009) further argues that though engagement with context risks a degree of syncretism, this is necessary because it is only through this that conversions become locally and meaningfully grounded.

Theologians such as John Mbiti and Emmanuel Idowu called for the indigenization of Christianity, while Lamin Sanneh and Kwame Bediako clamored for the translation of the Christian faith so that it would become more contextual. In a nutshell, they were arguing for the Africanization of the Christian faith while Christianizing African forms of belief. In fact, Africans questioned the demonization of African religious practices, yet adopting Western ones. Their questions were valid. For example, why would Western religious forms be regarded as Christian, while at the same time rejecting African religious forms? Did this mean that Western religious forms were synonymous with biblical Christianity? If so, why would African religious forms not also be regarded as biblical Christianity? Such questions aimed at decolonizing the Christian faith so that it could be understood through African

idioms. Idowu (1965), for example, called for the church in Nigeria (and throughout Africa) to be indigenized so that the expressions of Christianity would be performed through indigenous forms. In this case, there was no need for African Christians to sing foreign songs when in church or to dress like Europeans. He argued that African drums should be used in churches in order to make Christianity more authentically African. Mbiti (1969, p. 271), like Idowu, advocated for an indigenous African Christianity because, for him, it "holds the greatest potentialities of meeting the dilemmas and challenges of modern Africa". For Bediako (1992, p. 252), it is important for Christianity to be authentically African because it meets people in their context and communities, and transforms them within and from that setting. Commenting on the above theologians, Tarus and Lowery (2017, p. 313) opine that they "defended the need for Christians to thoroughly contextualize their faith, and insisted that the gospel was by its very nature both universal and particular it was intended to be 'translated' into each context in which it found itself". The birth of African theology, therefore, needs to be understood as a quest to make Christianity more relevant to African Christians. In order to do this, certain aspects of African Traditional Religion(s) were incorporated into churches. The emergence of African Indigenous Churches (AICs) gave impetus to this quest. They challenged the hegemony of Western Christianity by appropriating indigenous religious practices into their church rituals. Mbiti (1969, p. 268) postulated that traditional beliefs and practices would continue to thrive, even in towns and cities, for generations to come. He argued "any appeal made to traditional values and practices is ultimately a religious appeal. So long as people appreciate and even idolize the traditional present and past, this religiosity whether recognized as such or not will continue to enjoy a comfortable and privileged place in the emotions of African people". Mbiti's analysis is crucial for this paper because it assists us in understanding the prevalence of African indigenous religious beliefs and practices within PPMs in Zimbabwe. He raises a key point, which is discussed below. For example, that the introduction of a new religion does not mean the abandonment of the old. This is noted by Falconer (2018, p. 104) when he argues that "although society changes and religious beliefs and practices are transformed to suit new lifestyles, many Africans are not entirely detached from their traditional culture and worldview". Hence, Mbiti's argument is that the Christian faith is translatable, not only into different languages, but also into different cultural forms. For him, not Africanizing Christianity would lead to the dechristianization of Africa, because the religion would have remained foreign to Africans. It is not surprising, therefore, that within most Christian traditions (Pentecostalism included), in Africa in general and Zimbabwe in particular, aspects of African Indigenous Religion(s) are prevalent, either visibly or latent. This is what scholarship has termed 'syncretism'. Hollewerger (n.d.) opines that it cannot be questioned that all the different forms of Pentecostalism are syncretic. The section below theorizes syncretism in African Pentecostalism.

### 3. African Pentecostalism and Syncretism: Christianity Encounters ATR(s)

The issue of Pentecostalism and its syncretistic relationship with African Indigenous Religion(s) is not clearly defined (Falconer 2018, p. 104). In Falconer's analysis, this matter is complex, particularly when all of the expressions of belief are so diverse. However, the prevalence of syncretism in African Pentecostalism cannot go unnoticed. According to Anderson (2018, p. 190), when Pentecostalism entered Africa, it was inevitable that it would interact and form a dialogue with the old religion. Anderson noticed that thisin-teraction shows both continuity and discontinuity, as some aspects of African Indigenous Religion(s) are embraced while others are thrown away. Anderson (2001) notes that many observers regard as 'syncretistic' many forms of African Pentecostalism that have developed a pneumatology, with a presumed link to the pre-Christian past. Hence, by definition, African Pentecostalism is seen as referring to the distinctive modes of being Pentecostal that have come about as a result of processing and assimilating some African religious and cultural values, including the significance of the dead (Nel 2019, n.p). In his study of Iringa, Tanzania, Lindhardt (2017, p. 36) established that expressions of Pentecostal-Charismatic

Christianity have taken shape through an intimate and complex entanglement with African Indigenous Religion(s). In his opinion, African Pentecostalism has to adapt to African Indigenous Religions. Nel (2019) opines that African Pentecostalism accepts that African Indigenous Religion(s) provide a certain contact point or meeting place for communicating the gospel, and that African religiosity provides a religious groundwork, vocabulary, insights, aspirations and direction for the Christian gospel. Kalu (2008, p. 170) engages with a "cultural discourse that reconstructs the Pentecostal movement's response to the system of meanings embodied in the symbols and worldviews of indigenous African religions and cultures". He argues that [African] Pentecostalism has grown because it has resonated with indigenous worldviews and has responded contextually to questions that are raised within the interior of the worldviews. Kalu (2009, p. 71) further observes that "there is an identifiable African Pentecostalism because Africans responded to the gospel from within a charismatic indigenous worldview". In his analysis, Christianity acquired a different character as a result of African Pentecostalism, because it was now expressed in the idiom of the African world. Kalu's view is that African religions and cultures have contributed a specific flavor to African Pentecostalism. Hence, for him, the "conversation partners in shaping Pentecostal ideology and praxis are the indigenous religions and cultures among others" (Kalu 2008, p. 170). I, therefore, concur with Omenyo (2014, p. 132) when he says that African Pentecostalism oozes a certain Africanness. As a result of this, Nel (2019, n.p) argues that "in assimilating some elements of primal spirituality, African Pentecostalism can be suspected of syncretism that sacrifices the integrity of its proclamation of the gospel". However, Kalu (2008, p. 174) avers that the concept of syncretism seeks to respond to the persistent problem of Christ and culture. It is unfortunate, therefore, that Nel (2019) views the blending of some African indigenous religious practices with Christianity negatively. He does not clearly show, though, how the integrity of the way African Pentecostalism proclaims the gospel is sacrificed. His views were also shared by Kato (1975) when he spoke about syncretism becoming popular in third world countries, mainly due to the persistent urge for cultural revolution specifically in Africa, but with external influences from communist and Arab worlds. For him, this would energize the 'challenging' force of syncretism. Viewing syncretism as 'challenging' defeats the struggle for contextualizing the biblical message. In an era where the discourse on decoloniality is taking center stage, it is important for scholarship to desist from perpetuating negative descriptions of African Indigenous Religion(s). In the context of this study, I discuss syncretism in New Pentecostal Movements using an appreciative lens, requiring that religious movements deal with challenges emanating from the African worldview through a process of enculturation. For Lindhardt (2017, p. 36), through such a process, [African] Pentecostalism interacts, entangles, blends and borrows from African Indigenous Religion(s). Thus, Nwosu (2021, p. 12) is of the view that syncretism signifies equal and mutual borrowing. In his analysis, syncretism is a possible platform of collaborative working for justice and peace. Biri (2020) explains the resilience of indigenous beliefs and practices among the Shona people and argues that these have found expression among Zimbabwean Pentecostal churches. In the next section, I discuss syncretism in New Pentecostal Movements in Zimbabwe.

## 4. Prophetic Pentecostalism in Zimbabwe and Syncretism: When Faith Is Not Enough?

Pentecostalism in general is known for its emphasis on faith. Hebrews 11 is a key text for this Christian tradition. All the other aspects center on faith. Healing, deliverance and prosperity are all anchored on faith. According to Heb 11:1, faith is believing in what one hopes for and the evidence of those things that are not visible. Members of classical and Neo-Pentecostal churches are encouraged to believe in order for them to receive. As a way of breaking away from the past, they were discouraged from focusing on people or things as solutions to their troubles, but to trust in God. For a long time, members in these churches professed their faith in a God who heals, delivers and provides for their needs. In fact, they were taught that God answers prayers said with faith. Testimonies were given indicating that the God of the Pentecostal believer was at work as long as the believer had

faith. It would appear that such pronouncements are useful when a nation's economy is in good shape. Once the economy stumbles, evidence from Zimbabwe shows that faith alone may not be enough. Sande (2021) argues that it is one thing for Pentecostal leaders to claim to be able to heal and deliver people, and another to follow through in times of need. Zimbabwe provides a good case study of how the dynamics within African Pentecostalism had to change in order to suit the demands of the socio-political situation. The emergence of PPMs in Zimbabwe needs to be understood within this context.

Prophetic Pentecostalism arose in Zimbabwe during the crisis years. Within South Africa, Matshobane notes that these churches emerged within a context of high unemployment rates. This can also be said of PPMs in East, Central and West Africa. It needs to be understood that by nature, Africans are notoriously religious (Mbiti 1969). As a result, they seek for answers for any of life's challenges from their religious worldview. In Zimbabwe, the Prophetic Pentecostal Movements (PPMs) emerged in a context of economic and political challenges. The Zimbabwean environment was characterized by hyper-inflation, unemployment and political violence. Most scholars writing on Zimbabwean Pentecostalism have attributed the emergence of Prophetic Pentecostalism to the deteriorating economy. In a context where hope was rapidly fading away, these churches provided spaces for people to hope again by promising that things would become better in the future. Chitando (2021) argues that African Pentecostalism has thrived in Zimbabwe because of its ability to conform itself to the needs of the people in a variety of ways during the difficult times of the modern state. In Manyonganise's (2016, p. 269) opinion, PPMs were able to do this by moving from being 'other-worldly' to 'this worldly' as they endeavored to deal with poverty, violence and unemployment as well as other social pressures. In her analysis, the emergence of PPMs in Zimbabwe needs to be "understood as an attempt to offer people a way out of the cage of among other things the socio-economic and political uncertainties". As a result, the emergence of PPMs at the height of the Zimbabwean crisis from the year 2008 to the present, has challenged Zimbabwean Pentecostal Christians' sole dependency on faith. The crisis called for much more than faith could stand on its own. Hence, NPMs responded to this need by infusing some of the indigenous religious practices with biblical ones as a way of strengthening believers as they journeyed through the crisis.

From the onset, Prophetic Pentecostalism in Zimbabwe sought to move people away from their sole dependency on an abstract God by providing a leading visible figure who would assume some sacredness, just like in ATR(s) where sacred practitioners provided a rallying point in times of crisis or calamity. Such figures provided a link between the living and the ancestors. While earlier forms of Pentecostalism had leaders with titles such as 'pastors', 'teachers' and 'evangelists', founders of Prophetic Pentecostal Movements have the boldness of claiming titles like 'prophet' or 'seer', thereby sacralizing their persona (Biri 2021, p. 26). In order to authenticate their 'prophetic calling', these leaders have embarked on making prophecies (ranging from prophesying about one's cellphone number, home addresses and identity document numbers, to mention but a few). Claims of healing and deliverance miracles have been abundant, but a few have been authenticated. Throughout the years, these 'prophets' have been laughed off, yet communities hold them in high esteem. The reason is that they have so many affinities with indigenous sacred practitioners. As seer or prophet, just like the indigenous sacred practitioner, they claim that they are able to communicate with the spirit world to diagnose the problems facing members of their churches. Asamoah-Gyadu (2012) notes that in African Pentecostal Churches, the process of diagnosis has been associated with the work of prophets. Kgatle (2023) avers that problem diagnosis is one of the PPMs' practices that has made it popular. In ATR(s), it has always been the duty of a traditional diviner to diagnose people's problems and/or, at times, prescribe remedies in the form of medicines or rituals to be performed. The belief in witchcraft and sorcery is very strong in PPMs. While classical and Neo-Pentecostal movements emphasized the need to offer prayers for protection, Biri and Manyonganise (2022) note a new theology of retribution in PPMs where they send back the witchcraft curse to the sender. Chitando (2021, p. 14) explains that this practice is informed by the

indigenous worldview where the traditional healer can cause those who intend to harm others to endure the pain they intended to cause. In a Christian tradition that teaches forgiveness, Biri and Manyonganise (2022) question whether this is not reverse witchcraft.

Prophetic Pentecostal Movements (PPMs) in Zimbabwe introduced touchable objects or artefacts. Kgatle (2023) opines that these objects are prescriptions for those consulting the prophets in these movements. Manyonganise (2021, p. 94) notes that

> In UFIC and PHD, the selling of branded merchandise is very common. Among these are stickers, wristbands, posters and other paraphernalia. Both churches make use of branded 'annointing' oil. In fact, the annonting oil bottles have photographs of [the founders].

While this appears to be sophisticated syncretism, a critical analysis of the practices show how steeped they are in the African indigenous religious worldview. Biri (2021, p. 27) argues

> . . .wristbands and other forms of inscription that are a mark of identity, a new identity that have acquired by denouncing the past. Therefore, the two churches reinvent the traditional symbols and give them new meaning within the Christian set-up in order to attract clientele. This is a mark of genius by the two young leaders because most believers are familiar with the artefacts that they get from the traditional sacred practitioners, particularly the n'anga. UFIC and PHD give new meanings to their artefacts, but the appropriation of these traditional symbols defies the myth of making a complete break from the past as the Pentecostal ideology seeks to maintain. Hence, this is the revitalization of the indigenous beliefs and practices within the Pentecostal matrix.

Shoko and Chiwara (2013) equate such artefacts with the traditional sacred practitioner's charms. Manyonganise (2021, p. 95) indicates that believers in UFIC and PHD, as in other PPMs in Zimbabwe, attribute mythical powers to these artefacts, which they believe provide protection from harm caused by enemies and evil powers. From the believers' point of view, the prophets' 'anointing' resides in the artefacts. The same view is shared on the sacred practitioners' charms, which are thought to carry powers that protect people from harm, particularly from witchcraft and sorcery.

Scholarship focusing on African Pentecostalism has noted the emergence of new doctrines and ritual practices in Prophetic Pentecostalism. For example, the doctrine on fatherhood is very strong in these churches. It is a phenomenon that had not been witnessed in previous forms of Pentecostalism in the region in general and Zimbabwe in particular. At its onset, most prophets in southern Africa had spiritual fathers from West Africa. Questions may be raised pertaining to why this was/is the case. Chitando et al. (2013) tried to explain this in terms of young masculinities submitting themselves to mature or older masculinities, while Biri and Manyonganise (2022), focusing on witchcraft and the 'back to sender' phenomenon, explained the link between self-proclaimed prophets in southern Africa and their West African spiritual fathers as emanating from the knowledge of West Africa as the 'center' of ritual performances, some of which are rooted in African Indigenous Religions and are meant to give these 'prophets' power to make them famous through the performance of miracles. Dube (2017) locates the concept of 'fatherhood', which he calls 'spiritual parenting', within the African religious worldview. He argues that we need to understand this concept as providing alternative 'fatherhood' spaces, thereby implicitly reinventing traditional hegemonic models under the pretext of Christian spiritualities. In his analysis, the spiritual parent ensures spiritual protection of the 'child', a similar role played by the traditional father. In the same vein, when analyzing UFIC, Biri (2021) is of the view that its founder, Emmanuel Makandiwa, is cast as the father of the unified family, which, to a large extent, has enabled members in his church to break away from their birth families and assume a new identity in this new family. While Biri's analysis may be true, she may have failed to see beyond Makandiwa's love for family. For example, in most of his teachings, he has called on members of his church to cherish their families of birth, particularly their biological parents and guardians who raised them. In order to buttress

this point, he has ensured that his parents and in-laws, who are members in his church, are well-looked after as an example to his congregants. His siblings constitute the group of those who are called to ministry and play a significant role in his church. This has led critics to point out that he is building a dynasty similar to traditional modes of kingship and kinship.

The concept of guesthouses is also a new trend which exhibit African Pentecostal borrowings from ATR(s). In Zimbabwe, prophets like Emmanuel Makandiwa (UFIC) and Walter Magaya (PHD) have established guesthouses where people pay to meet the prophet one-on-one. Earlier forms of Pentecostalism prayed for the sick for free and the pastors would visit the sick in their homes. Prophetic Pentecostalism has departed from this practice. It is the sick who now visit the prophet at the guesthouses, but for a 'fee'. Such practices have been criticized for being selective as the poor cannot afford the boarding fees demanded at these guesthouses. Hence, the practice is seen as elitist, that is, catering only for the well-to-do in society. Manyonganise (2020) criticized the concept for not only extorting money from the unsuspecting public, but also for commoditizing spiritual or faith healing, thereby excluding the poor and vulnerable. Her tools of analysis were rooted in the understanding that faith healing in Pentecostal churches should be free. A question that quickly comes to mind is: why should faith healing be free in Pentecostal churches when other religions charge for their services? A critical analysis of this practice shows its prevalence in ATR(s). Manyonganise (2020) argues that the introduction of guesthouses by the founders of UFIC and PHD is not a novel development, since even African indigenous health practitioners had/have the tendency of establishing their own healing centers. Among the Shona, such centers are referred to as '*banya*' (healing house) and these were usually places constructed away from homesteads. It therefore becomes clear that the PPMs have just modernized the practice. Biri (2012, p. 42) acknowledges the ability of African Pentecostalism to innovate indigenous religious practices by resacralizing, reinterpreting and redefining them.

In southern Africa in general and Zimbabwe specifically, Prophetic Pentecostal Movements have also brought in another new trend, that is, the concept of seeding. This, too, is a practice rooted in ATR(s). For example, when approaching a traditional diviner, herbalist or healer in ATR(s), it was/is expected that one had/has to carry some form of payment or appreciation gift so that a blessing can be pronounced upon them. The general belief is that once this indigenous practitioner is appeased, they are bound to do all they can to ensure that good fortune comes the way of the person approaching them. This forms the basis of the PPMs' prosperity gospel, which resonates very well with African indigenous rituals of prosperity. Gadsby (2022, p. 126) notes that ATR(s) are replete with rituals that convey the belief that personal prosperity is a sign of the approval of the spirit world. While giving for blessings has always been part of the Pentecostal doctrine, the PPMs have taken it to another level. Citing 1 Samuel 9, these prophets have claimed that anyone in need of their help should not approach them empty-handed. At times, the well-to-do in these churches are given prophecies to the effect that God has commanded them to give a certain amount of money to the prophets in order for them to make 'breakthroughs' in their lives. This has also led to the creation of tiers within the churches whereby spiritual partners are put into classes depending on how much they pay per month to support 'the work of God'. Those who pay are provided with seats close to the prophet, which is deemed 'close to the anointing'. Hence, they are referred to as Very Very Important Persons (VVIPs). What this implies is that one's relationship with God is transactional. Hence, while a lack of giving and tithing were given as reasons for lack of financial breakthroughs in earlier forms of Pentecostalism, Prophetic Pentecostalism has placed the persona of the prophet at the center of one being blessed or cursed. It is common, therefore, to hear followers of prophets in Zimbabwe saying '*nyasha dzemuporofita dzive nemi*' (the grace of the prophet be with you). God suddenly becomes the God of the prophet. As they pray, church members can be heard saying 'the God of Prophet so and so'. As such, we can argue that the figure of the prophet in these churches is slowly replacing that of Jesus Christ. In fact, a new

hierarchy is slowly coming into place where the prophet is deemed to be an intermediary between believers and Jesus, yet other forms of Pentecostalism hold the belief that anyone can freely access the throne of God. Kgatle (2023) notes that in PPMs, the prophet is more important than the participation of all believers. In order to maintain the figure of the prophet as mysterious, access to the prophet is not easy and they are always surrounded by bodyguards. Shoko and Chiwara compares this to traditional sacred practitioners who had their own assistants who would ensure that one is either granted or denied access to appear in the presence of the diviner, healer or herbalist. In PPMs, it is this inaccessibility that leads followers to part with hard-earned cash to go for one-on-one meetings with the prophet at the established guesthouses. While this may reflect as 'purely' Christian, it has its roots within ATR(s) where the sacred practitioners is deemed to be the intermediary between the living and the ancestors. Manyonganise (2021) describes this as rebranding Pentecostalism, and Chitando (2021) also views PPMs as rebranding ATR(s). Gadsby (2022, p. 125) notes that the multifarious rebranding of ATR(s) has been key to the success of UFIC and PHD, and the leader of each has transformed the traditions strikingly successfully.

## 5. African Pentecostalism and the Resilience of ATR(s): Towards a Decolonization of Religion in Africa

The discussion above shows the resilience of indigenous religious practices within a Christian tradition that claims to have totally broken from the past. Therefore, the fast growth of PPMs depends on the 'Christianization' of indigenous religious practices, which are presented to believers as 'purely biblical'. The reference to 'biblical' needs further interrogation. Manyonganise (2023), while acknowledging the centrality of the written Bible in Africa, also makes reference to the 'bible of culture'. Both are bibles, but emerging from different contexts. The latter is critical in indigenizing Pentecostalism in Africa. Kgatle (2023) views this as contributing to the decolonization discourse in Africa. He argues that the indigenization of the Pentecostal movement is relevant to the decolonization of the Western expression of Christianity. In his analysis, the indigenized and contextualized gospel in Africa means that Africans are given an opportunity to drink from their own wells. In this case, the syncretic nature of PPMs needs to be viewed positively as a way of trying to understand Christianity using indigenous resources that are useful for meaning-making in both Pentecostal and African spiritualities. Chitando (2021, p. 9) argues that African Pentecostalism must be prepared to engage with ATR(s). If African scholarship is to make a meaningful contribution to the decolonization discourse, they need not rely on theories invented elsewhere to analyze the cultural encounters between African Pentecostalism and African Traditional Religion(s). Relying on analyses by scholars such as Sundkler would only result in syncretism being viewed negatively. Mokhoathi (2019) casts aspersions on the detached approaches that Western scholars have used in the interrogation and evaluation of the African religious heritage. It is critical, therefore, for African scholars to have the courage to develop their own tools of analysis that challenge such views and justify the necessity of such cultural encounters. This is useful in decolonizing religion on the African continent, as it makes Christianity a truly African religion. Kgatle and Mofokeng (2019) call for the affirmation of the Africanness and the Christianness of the African Pentecostal community as significant components in nurturing a critical hermeneutic of experience. African scholarship on religion must, therefore, query the continued labelling of African indigenous religious beliefs and practices as evil in need of being thrown into the dustbins of history. The resilience of such practices attests to the fact that they are alive and thriving in the hearts of Africans. In this case, PPMs in Africa in general and Zimbabwe in particular can be viewed as a useful resource for the decolonization process.

## 6. Conclusions

The intention of this chapter was to examine the encounters between Prophetic Pentecostalism and African Traditional Religion(s) in Zimbabwe. Such encounters were shown to be located within the discourse on syncretism. In order to situate the subject under

discussion in its proper context, a theorization of syncretism was undertaken, highlighting its positive aspects as well as the challenges associated with the term. This article then went on to examine syncretism in African Pentecostalism in general before it engaged with the cultural encounters between PPMs and ATR(s) in Zimbabwe. An engagement with scholarship on African Pentecostalism showed that whenever two religions meet, there is bound to be borrowing, blending and reinterpretation. Having understood this, I, therefore, argued for a positive evaluation of the term and process of syncretism. An examination of cultural encounters between PPMs and ATR(s) in Zimbabwe lead us to the following conclusions. First, we can conclude that lending and borrowing at the confluence of cultural encounters cannot be avoided. Hence, instead of viewing these as negative, it is important to appreciate them. Second, practices and beliefs in PPMs in Zimbabwe show that they have adopted and adapted quite a number of African indigenous practices and beliefs within their own theology. Third, a decolonialized discourse requires that religion plays a significant role not only in its own decolonization, but in the decolonization of other aspects of society. For example, Christianity, which colluded with colonial powers, needs to atone itself of this label by engaging meaningfully in the decolonization process. What this implies is that it cannot continue to demonize ATR(s), but should encourage the appropriation of positive aspects of the religion not only in PPMs, but other Christian traditions as well. This has largely been a desktop research project; future research on the subject requires that empirical data be presented in order to show the attitudes of members of PPMs in Zimbabwe towards the appropriation of African religious beliefs, practices and symbols within their Pentecostal practices.

**Funding:** This research received no external funding.

**Institutional Review Board Statement:** Not Applicable.

**Informed Consent Statement:** Not Applicable.

**Data Availability Statement:** No new data were created or analyzed in this study. Data sharing is not applicable.

**Conflicts of Interest:** The author declares no conflict of interest.

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
