# Peer review of "‘When Faith Is Not Enough’: Encounters between African Indigenous Religious Practices and Prophetic Pentecostal Movements in Zimbabwe"

_religions, doi:10.3390/rel15010115_

Round 1
Reviewer 1 Report
Comments and Suggestions for Authors
A well processed and well written manuscript with potential.
This work promises to be a valuable contribution to scholarship in the broader Prophetic Pentecostal Movement not only in Zimbabwe but continent of Africa.
I wondered if you considered the similarities of some of the syncretic practices observed in leadership/ministry styles of African Prophets to those of their counterparts in the West or elsewhere.
To what extend are those practices influenced/informed by similar practices of the founding missionaries or contemporary Prophets in the sending missions?
A brief discussion on this phenomenon may be helpful, only, if considered necessary.
Author Response
Responses have been attached

Reviewer 2 Report
Comments and Suggestions for Authors
The article presents a desktop study of the 'religious encounters between indigenous African religious practices and African Prophetic Pentecostal Movements (PPMs) in the Zimbabwean context. The author provides the socio-economic context in which the PPMs arose, including the inadequacy of the standard Christian approaches to felt needs. The attraction of PPMs is located in their adoption of indigenous religious practices, both in the role PPMs' prophets assume in relation to African traditional healers and the tangible objects they use. The author acknowledges this adoption as syncretistic and argues for a positive appraisal of syncretism, firstly because a non-syncretic religion is historically unimaginable. Secondly, the demands of decolonisation necessitate a wariness against Christianity as received from the West and a championing of homebrewed faith.
The document has some errors highlighted for the author's attention. Otherwise, the article is well argued and represents an important contribution on the subject of syncretism and the growing phenomena of Prophetic Pentecostalism.

The quality of the English language is acceptable. Minor issues have been highlighted in the document.
Author Response
Responses have been attached
